# TS-LLaVA: Constructing Visual Tokens through Thumbnail-and-Sampling for Training-Free Video Large Language Models

## Abstract

Recent advances in multimodal Large Language Models (LLMs) have shown great success in understanding multimodal contents. For video understanding tasks, training-based video LLMs are difficult to build due to the scarcity of high-quality, curated video-text paired data. In contrast, paired image-text data are much easier to obtain, and there is substantial similarity between images and videos. Consequently, extending image LLMs for video understanding tasks presents an appealing alternative. Developing effective strategies for compressing visual tokens from multiple frames is a promising way to leverage the powerful pre-trained image LLM. In this work, we explore the limitations of the existing compression strategies for building a training-free video LLM. The findings lead to our method TS-LLaVA, which constructs visual tokens through a Thumbnail-and-Sampling strategy. Given a video, we select few equidistant frames from all input frames to construct a Thumbnail image as a detailed visual cue, complemented by Sampled visual tokens from all input frames. Our method establishes the new state-of-the-art performance among training-free video LLMs on various benchmarks. Notably, our 34B model outperforms GPT-4V on the MVBench benchmark, and achieves performance comparable to the 72B training-based video LLM, Video-LLaMA2, on the challenging MLVU benchmark. We will release code upon acceptance.

## 1 Introduction

Thanks to the breakthroughs in foundation models like CLIP (Radford et al., 2021) and Large Language Models (LLMs) (Touvron et al., 2023; Bai et al., 2023a; Jiang et al., 2024; Chiang et al., 2023), image LLMs (Zhang et al., 2024c; Bai et al., 2023b; Liu et al., 2023; 2024a) have shown great success in comprehending visual contents and generating textual responses based on user prompts. Image LLMs are often trained with paired image-text data to bridge the representation spaces between the vision encoder and the LLM. Although trained in a similar way, video LLMs (Maaz et al., 2024; Lin et al., 2023; Xu et al., 2024a; Li et al., 2024c) require far more resources and a larger volume of paired video-text data due to the increased complexity of video content. Unfortunately, curated large-scale video-text data is hard to obtain. Since images are easier to acquire and share similar features with videos, extending image LLMs for video understanding presents a promising approach.

We illustrate the training-free video LLM in Figure 1. To bypass the high computational cost in training-based video LLM, several methods have been proposed that use pre-trained image LLMs to build training-free video LLMs (Wu, 2024; Xu et al., 2024b; Kim et al., 2024). Given an image LLM, the vision tower directly encodes the spatial information of each frame, while the autoregressive nature of the LLM captures the temporal information within the input sequence to some extent. However, the number of visual tokens[1] generated from videos can be much larger than those from images, which is problematic given the limited context length of the pre-trained image LLMs. Consequently, a key design aspect of training-free video LLM is the visual token compression strategy, which enables image LLMs to efficiently handle video input.

---

[1]i.e. elements of the encoded visual features, used as input for the LLM.

**Training-free video LLM**

Training:
Inference:

Trained on **image-text data only**          Inference on **video-text data**

Figure 1: Illustration of training-free video LLM. Vision Tower: vision encoder and projection module in image LLM.

To better understand the working mechanisms of training-free video LLM, we begin by evaluating the commonly used compression strategies. Insights gained from examining these visual token compression strategies led to the development of our method, TS-LLaVA. TS-LLaVA adopts a hybrid manner to compress visual tokens. Given $N$ input frames, we select $N_T$ ($N_T \ll N$) frames to construct a **T**humbnail image that serves as a summary of the video. We show that using this thumbnail image alone enables the model to comprehend certain aspects of the video contents, though it still leaves some nuances unaddressed. To address this, we complement the thumbnail image by incorporating **S**ampled visual tokens from all frames, which leads to a deeper and more complete grasp of the video content. Together, the **T**humbnail-and-**S**ampling strategy constitutes the core of our TS-LLaVA.

We evaluate our method on various video understanding benchmarks. Experimental results show that TS-LLaVA outperforms the previous state-of-the-art (SOTA) training-free video LLMs by a large margin. Notably, our 34B model surpasses GPT-4V (OpenAI, 2023a) on the MVBench benchmark (Li et al., 2024b), and achieves performance comparable to the 72B training-based video LLM Video-LLaMA2 (Cheng et al., 2024) on the challenging MLVU benchmark (Zhou et al., 2024). We conduct comprehensive ablation studies to assess the design choices of TS-LLaVA. Together with the study on compression strategies, we hope to provide insights for future works. In summary, our main contributions include:

- We conduct a thorough evaluation of various compression strategies for training-free video LLMs, offering insights to guide future research in developing video LLMs.

- Based on our findings, we propose the Thumbnail-and-Sampling compression strategy, which shows clear advantages over existing approaches.

- Leveraging the Thumbnail-and-Sampling strategy, we develop TS-LLaVA, establishing a new SOTA among training-free video LLMs while maintaining high token efficiency.

## 2 Related Works

**Image LLMs** aim to bridge the representation space between the vision encoder (Radford et al., 2021) and LLM (Touvron et al., 2023; Bai et al., 2023a; Jiang et al., 2024; Chiang et al., 2023) using image-text data. The BLIP family, including BLIP-2 (Li et al., 2023b) and InstructBLIP (Dai et al., 2023), employs a Querying

Transformer (QFormer) to connect vision and language modalities via learnable queries in cross-attention modules. QFormer is also adopted by follow-up works, such as Qwen-VL (Bai et al., 2023b) and mPLUG-Owl (Ye et al., 2023). Qwen-VL incorporates interleaved image-text data in a three-stage training pipeline, while mPLUG-Owl adopts a modularized learning approach. The LLaVA family, in contrast, uses an MLP for connecting the vision encoder and LLM. LLaVA (Liu et al., 2023) leverages GPT-4 (OpenAI, 2023b) generated visual instruction data for fine-tuning. Subsequent versions, LLaVA-v1.5 (Liu et al., 2024a) and LLaVA-v1.6(NeXT) (Liu et al., 2024b) further improve LLaVA's performance through better data, higher resolution and stronger LLM. As a concurrent work, MiniGPT-4 (Zhu et al., 2023) aligns a frozen LLM with frozen ViT (Dosovitskiy et al., 2021) and QFormer using a trainable linear layer to enable multimodal instruction following.

**Training-based video LLMs** are further trained on massive video data to endow image LLMs or LLMs with video understanding capabilities. Video-ChatGPT (Maaz et al., 2024) uses LLaVA as its backbone, applying separate temporal and spatial pooling to visual features. The model further utilizes 100k video instruction tuning data. VideoChat (Li et al., 2024a) leverages QFormer for video token compression, and VideoChat2 (Li et al., 2024b) refines this approach with improved vision-language alignment and instruction tuning. It also introduces the multitask video understanding benchmark, MVBench. Video-LLaVA (Lin et al., 2023) learns a shared projector for image and video encoders. Video-LLaMA (Zhang et al., 2023) and Video-LLaMA2 (Cheng et al., 2024) incorporate video, audio and language modalities to support tasks oriented toward video and audio. LLaVA-NeXT-Video (Zhang et al., 2024d) fine-tunes LLaVA-NeXT on video data, with a variant that applies DPO (Rafailov et al., 2023) for improved performance. LITA (Huang et al., 2024) employs a slow-fast design (Feichtenhofer et al., 2019; Xiao et al., 2020) to capture spatial and temporal information more effectively.

**Training-free video LLMs** extend image LLMs for video understanding without requiring additional fine-tuning on video data. As a pioneering approach, IG-VLM (Kim et al., 2024) constructs a grid-view image from video frames, which is then fed directly into a frozen image LLM with a specially designed prompt. While promising, the image grid approach has limitations, such as reduced resolution and the limited number of frames it can include, which we further discuss in the next section. FreeVA (Wu, 2024) explores various temporal aggregation methods, but similarly uses a limited number of frames. The current state-of-the-art SF-LLaVA (Xu et al., 2024b) adopts the slow-fast design, which is proven to be effective in action recognition (Feichtenhofer et al., 2019; Xiao et al., 2020), and in LITA, as mentioned earlier. SF-LLaVA designs a slow pathway compressing fewer frames, and a fast pathway heavily compressing more frames. While both SF-LLaVA and our method use a two-stream design, our Thumbnail-and-Sampling strategy leads to significantly better performance on various video understanding benchmarks, while maintaining a better token efficiency. We extend the discussion in Section 4.

## 3 Method

### 3.1 Comparing Different Compression Strategies

To equip the baseline image LLM with the video understanding abilities, various visual token compression strategies are applied. They allow incorporating multiple frames from the videos. Including commonly used methods, we evaluate five types of compression strategies as follows:

- *Concat*: Directly concatenating frames (no compression).

- *Pooling*: Applying spatial average pooling to visual tokens.[2]

- *Grid*: Creating a single grid-view image composed by $N_T$ frames to represent a video.

- *Grids*: For $N = k \times N_T$ frames, generating $k$ grid-view images, each composed of $N_T$ frames.

- *Sampling*: Applying uniform sampling to visual tokens.

---

[2]Different pooling operations are compared in the Appendix.

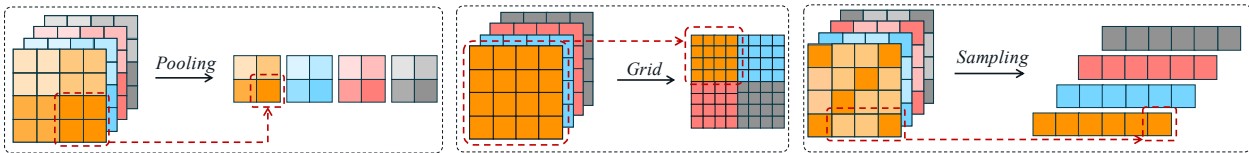

Figure 2: Visual token compression strategies illustrated. *Pooling* and *Sampling* operate on encoded tokens, *Grid* operates on RGB images. We omit the encoding procedure for simplicity. We extend *Grid* to *Grids* by composing multiple grid view images.

Specifically, *Concat* is one of the core ideas of the training-free video LLM FreeVA (Wu, 2024). *Pooling* is a widely used token compression method for video LLMs (Li et al., 2024c; Cheng et al., 2024; Huang et al., 2024; Maaz et al., 2024; Xu et al., 2024a; Zhang et al., 2024b), including training-free methods (Wu, 2024; Xu et al., 2024b). *Grid*, as proposed by IG-VLM (Kim et al., 2024), aims to transform video frames into a grid image for better interaction with image LLMs. We extend this to *Grids* by composing multiple grid-view images. Finally, we also evaluate *Sampling* as a less explored compression strategy for video LLMs. Figure 2 illustrates the key components of these strategies.

We evaluate different visual token compression strategies on the Multiple Choice VideoQA task. Unlike Open-Ended VideoQA, the evaluation of Multiple Choice VideoQA does not require an additional language model. We adopt Video-MME (Fu et al., 2024) for evaluation, as it categorizes videos by duration (short, medium, and long) as well as by task type, providing a more comprehensive assessment of each compression strategy's performance. We use LLaVA-v1.6-7B (Liu et al., 2024b) (Vicuna-v1.5 version) as the backbone image LLM, with a context length of 4096 tokens. Following standard protocol, input frames are uniformly sampled from each video. To ensure fair comparison, we set the number of visual tokens to 2304 after compression (except for "*Grid*", where only one grid-view image is used, totaling 576 tokens). The compression rate, calculated as the ratio of visual tokens before to after compression, is set to 4.

We present the results in Table 1. The simplest approach to adapting an image LLM for video understanding is to concatenate frames (*Concat*). While *Concat* avoids compressing visual tokens, it is limited by the number of frames it can process. *Concat* is considered as the baseline method, which shows how powerful is the image LLM for video understanding tasks without any modification.

Surprisingly, despite being widely used, *Pooling* does not demonstrate superior performance compared to other methods. With 16 frames, *Pooling* only slightly outperforms the baseline *Concat* (4 frames) in overall accuracy, but shows worse reasoning abilities than *Concat*.

Composing grid-view images (*Grid*, *Grids*) yields promising results. *Grid* can be seen as using a thumbnail image to represent the video. Even with only

Table 1: Comparison between different compression strategies on Video-MME dataset: Results obtained using LLaVA-v1.6-7B. "Vis. Tokens": Visual Tokens; "Comp.": Compression.

| Compre. Strategy | #. of Frames ($N$) | #.of Vis. Tokens | Comp. Rate | Video-MME | | | |
|---|---|---|---|---|---|---|---|
| | | | | Short | Medium | Long | Overall |
| *Concat* | 4 | 2304 | | 49.2 | 41.0 | 36.9 | 42.4 |
| *Pooling* | 16 | 2304 | 4 | 49.6 | 42.6 | 36.4 | 42.9 |
| *Grid* | 4 | 576 | 4 | 47.0 | 39.8 | 36.2 | 41.0 |
| *Grids* | 16 | 2304 | 4 | **52.2** | 42.0 | **37.6** | 43.9 |
| *Sampling* | 16 | 2304 | 4 | 52.0 | **43.0** | 37.3 | **44.1** |

(a) Overall performance (Accuracy)

| Compre. Strategy | #. of Frames ($N$) | #.of Vis. Tokens | Comp. Rate | Video-MME Reasoning Tasks | | | |
|---|---|---|---|---|---|---|---|
| | | | | Temporal | Spatial | Action | Object |
| *Concat* | 4 | 2304 | | 32.8 | 62.5 | 37.2 | 40.5 |
| *Pooling* | 16 | 2304 | 4 | 31.1 | 62.5 | 37.2 | 39.6 |
| *Grid* | 4 | 576 | 4 | 31.1 | 60.7 | 36.1 | 38.5 |
| *Grids* | 16 | 2304 | 4 | **34.5** | **67.9** | 37.2 | **40.7** |
| *Sampling* | 16 | 2304 | 4 | 33.3 | 66.1 | **38.9** | 40.5 |

(b) Performance (Accuracy) on reasoning tasks

1/4 of the tokens used by other methods, *Grid* achieves satisfactory performance, allowing for reasonable video comprehension. However, due to its limited number of frames and visual tokens, Grid is ultimately outperformed by other compression strategies. By using 16 frames to form 4 grid-view images (*Grids*), we significantly improve the performance of the baseline LLaVA-v1.6 (*Concat*) while maintaining the same

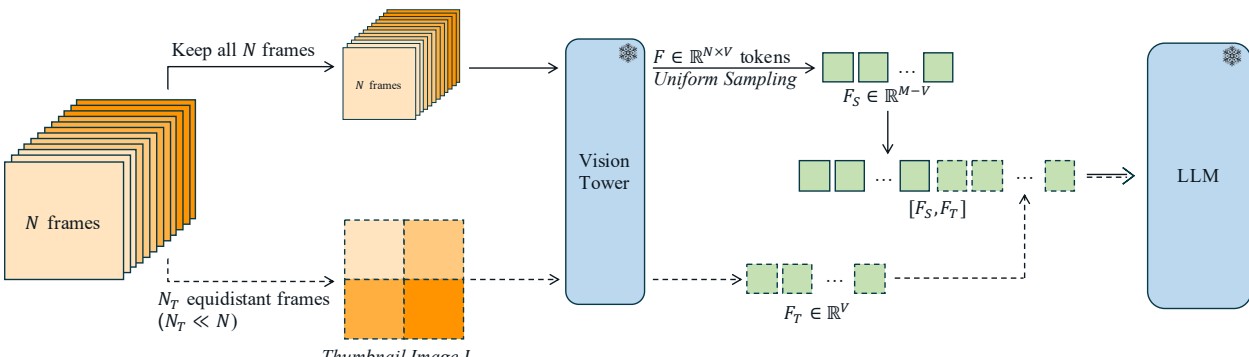

Figure 3: Illustration of our TS-LLaVA. The vision tower includes vision encoder and projection module in image LLM. The dashed lines and solid lines trace the procedures for constructing the thumbnail image tokens and sampled image tokens, respectively. $V$ denotes the number of visual tokens from the vision tower, and $M$ is the pre-defined number of visual tokens. We omit text input to LLM for simplicity.

visual token counts. Surprisingly, as a less commonly used compression strategy in video LLM, *Sampling* leads to the best overall performance comparing to other methods.

Moreover, as shown in Table 1b, composing multiple grid-view images (*Grids*) and uniformly sampling visual tokens (*Sampling*) shine on different types of tasks. They reach similar performance for object reasoning tasks. While *Grids* performs better on temporal and spatial reasoning tasks, but it falls short on action reasoning tasks. This outcome aligns with the characteristics of these methods. Directly sampling visual tokens may lead to missing details in some frames, while preserving them in others, potentially disrupting the temporal or spatial reasoning process. In contrast, composing grid-view images results in low-resolution frames, which could harm the model's ability to reason about actions effectively.

We summarize our findings as:

- Despite its popularity, *Pooling* does not yield satisfactory results for training-free video LLMs.

- *Grids* and *Sampling* show promising results, though each has certain limitations.

## 3.2 TS-LLaVA

In this section, we introduce TS-LLaVA and its core idea, the Thumbnail-and-Sampling strategy.

The experiment in Section 3.1 shows that both *Grids* and *Sampling* offer superior video understanding capabilities compared to other strategies. However, due to their respective designs, *Grids* is limited to low-resolution frames; while *Sampling* risks losing details from unsampled tokens.

*How to overcome the shortcomings of these two compression strategies?* Recall that *Grid*, a simplified version of *Grids*, performs well on reasoning tasks but is constrained by the limited number of frames and visual tokens it can process. To leverage the strengths of both approaches, we propose an intuitive method to combine their benefits, as illustrated in Figure 3. We select few equidistant frames from all input frames to create a grid-view image (*Grid*), which serves as the thumbnail image for the video. The missing information from this thumbnail is then complemented by sampling visual tokens from all input frames (*Sampling*) at the original resolution. This approach creates a more versatile compression strategy, effectively combining the strengths of both *Grids* and *Sampling*.

Specifically, given an input video, we first uniformly sample $N$ frames from the video. These frames are further used for video understanding with a given image LLM.

**To construct Thumbnail image tokens**, we further select $N_T$ equidistant frames from the initially obtained $N$ frames, where $N_T \ll N$ and $N_T \equiv 0 \pmod 2$. We create the thumbnail image by arranging

these frames into a single grid-view image, $I_T$, that accommodates all selected $N_T$ frames. $I_T$ is then processed by the vision encoder and projection module to generate thumbnail image tokens $F_T \in \mathbb{R}^V$, where $V$ denotes the number of visual tokens from the vision tower.

**To construct Sampled image tokens**, all $N$ frames are used. We first extract visual features frame by frame, resulting in features $F \in \mathbb{R}^{N \times V}$. Given a pre-defined maximum number of visual tokens $M$, we apply:

$$F \in \mathbb{R}^{N \times V} \xrightarrow{Sample} F_S \in \mathbb{R}^{M-V} \tag{1}$$

where *Sample* denotes uniform sampling, and $V < M <$ context length of LLM.

Finally, we obtain the visual tokens by concatenating $F_S$ and $F_T$ as $[F_S, F_T]$, which is further passed to LLM to interact with encoded textual inputs for video understanding.

### 3.3 Empirical Validation

To demonstrate the advantages of our Thumbnail-and-Sampling strategy, we conduct experiments on Video-MME under the same setting as in Section 3.1. From the 16 input frames, we select 4 (†) or 6 (‡) equidistant frames to compose the thumbnail image, resulting in 576 encoded visual tokens. All 16 input frames are encoded to extract the sampled visual tokens. We use 2304 visual tokens in total.

The results are presented in Table 2, where our method achieves the best overall performance. Our compression strategy brings notable improvements on both short and long videos, while maintaining the strong performance of *Sampling* or *Grid/Grids* on medium-length videos. Table 2b further shows that our compression strategy effectively leverages the strengths of *Sampling* and *Grid*. Compared to *Grids*, *Sampling* lags on spatial reasoning, while it excels on action reasoning. Our strategy improves the performance on these two tasks by a large margin, indicating that our strategy integrates the advantages of *Grid* and *Sampling*.

Moreover, although the overall compression rate remains the same, our method uses fewer sampled tokens compared to *Sampling* under the same setting (1728 vs 2304 tokens). Replacing 576 sampled

Table 2: Comparison between different compression strategies on Video-MME dataset using LLaVA-v1.6-7B. "*T-and-S*": Thumbnail-and-Sampling; "Vis. Tokens": Visual Tokens; †/‡: using grid view (thumbnail) image composed of 4 or 6 frames.

| Compre. Strategy | #. of Frames ($N$) | #.of Vis. Tokens | Comp. Rate | Video-MME | | | |
|---|---|---|---|---|---|---|---|
| | | | | Short | Medium | Long | Overall |
| *Grid* | 4 | 576 | 4 | 47.0 | 39.8 | 36.2 | 41.0 |
| *Grids* | 16 | 2304 | 4 | 52.2 | 42.0 | 37.6 | 43.9 |
| *Sampling* | 16 | 2304 | 4 | 52.0 | **43.0** | 37.3 | 44.1 |
| *T-and-S*† | 16 | 2304 | 4 | **54.0** | 41.9 | **38.4** | 44.8 |
| *T-and-S*‡ | 16 | 2304 | 4 | 52.9 | 42.9 | 38.2 | 44.7 |

(a) Overall performance (Accuracy)

| Compre. Strategy | #. of Frames ($N$) | #.of Vis. Tokens | Comp. Rate | Video-MME Reasoning Tasks | | | |
|---|---|---|---|---|---|---|---|
| | | | | Temporal | Spatial | Action | Object |
| *Grid* | 4 | 576 | 4 | 31.1 | 60.7 | 36.1 | 38.5 |
| *Grids* | 16 | 2304 | 4 | **34.5** | 67.9 | 37.2 | 40.7 |
| *Sampling* | 16 | 2304 | 4 | 33.3 | 66.1 | 38.9 | 40.5 |
| *T-and-S*† | 16 | 2304 | 4 | 33.9 | **71.4** | 40.4 | **41.6** |
| *T-and-S*‡ | 16 | 2304 | 4 | **34.5** | 69.6 | **41.1** | 39.4 |

(b) Performance (Accuracy) on reasoning tasks

tokens with visual tokens from the thumbnail image clearly improves the accuracy of downstream tasks. The merits of our method are further discussed in the next section.

## 4 Experiments

### 4.1 Experimental Setups

**Multiple Choice VideoQA** Following Kim et al. (2024); Xu et al. (2024b), we evaluate our method on three Multiple Choice VideoQA benchmarks, including NExT-QA (Xiao et al., 2021), EgoSchema (Mangalam et al., 2023), and IntentQA (Li et al., 2023a). Following Kim et al. (2024), apart from the overall accuracy on each dataset, we report accuracies on Casual, Temporal and Descriptive tasks from NExT-QA dataset.

**Multitask Benchmarks** We also evaluate our method on two multitask video understanding benchmarks.

MVBench (Li et al., 2024b) includes 20 sub-tasks, spanning 9 types of reasoning abilities in video under-standing. For the classified categories included in the benchmark, please refer to the Appendix. The task types can be seen in Table 5.

MLVU (Zhou et al., 2024) is a multitask long video understanding (LVU) benchmark, which evaluates 3 types of capabilities of Video LLM. We conduct experiments on multiple choice questions, covering Holistic LVU, Single-Detail LVU and Multi-Detail LVU. For task types, please refer to Table 5. We report the results from the official evaluation server.

We use VideoMME (Fu et al., 2024) to evaluate the effectiveness of our compression strategy (Section 3). This benchmark categorizes videos into three duration-based types: short, medium, and long. We report results for our full model in this section.

LongVideoBench (Wu et al., 2024) focuses on long video understanding capabilities. It contains 3,763 videos with 6,678 human-annotated questions across 17 categories. The average duration per video is 473 seconds. We report results on its validation set.

**Notes on Open-Ended VideoQA** We also evaluate our method on Open-Ended VideoQA benchmarks and video-based Text Generation tasks. For Open-Ended VideoQA, we adopt MSVD-QA (Chen & Dolan, 2011), MSRVTT-QA (Xu et al., 2016), TGIF-QA (Li et al., 2016) and ActivityNet-QA (Yu et al., 2019). For Text Generation, we use VCGBench (Maaz et al., 2024). However, the GPT-assisted evaluation for these tasks is not as reliable as the evaluation protocol for Multiple Choice VideoQA. Although we reach the SOTA level performance on these benchmarks, we put the results in the Appendix. We urge for the developing of a better evaluation protocol on these benchmarks. For more discussion, please refer to Section B.2.

**Implementation Details** Following Kim et al. (2024); Xu et al. (2024b), we use LLaVA-v1.6 (Liu et al., 2024b) (7B and 34B) as our backbone image LLM. The 7B model adopts Vicuna-v1.5 (Chiang et al., 2023) as LLM, and 34B model uses Nous-Hermes-2-Yi-34B (NousResearch, 2023) as LLM. We conduct experiments on NVIDIA A100 80G GPUs. Following Xu et al. (2024b), we resize the input frames to 336×336. We use maximum 50 uniformly sampled frames per video, among which we select 6 equidistant frames to compose the thumbnail image. The encoded thumbnail image corresponds to 576 visual tokens. Then for the 50 input frames, we uniformly sample 2880 tokens from 50×576=28800 tokens. Together, we use 576+2880=3456 visual tokens, which is slightly lower than 3680 tokens used by SF-LLaVA (Xu et al., 2024b).

## 4.2 Main Results

### 4.2.1 Multiple Choice VideoQA

We first report the overall accuracy on Multiple Choice VideoQA tasks in Table 3. Among the training-free methods built on top of an open-source LLM, both our 7B and 34B models achieve SOTA performance across all three benchmarks. Notably, on the challenging EgoSchema dataset, which emphasizes long-form temporal reasoning in video LLMs, TS-LLaVA surpasses the previous SOTA SF-LLaVA by 3.0 and 2.0 percentage points for the 7B and 34B configurations, respectively. Furthermore, SF-LLaVA fails to outperform VideoTree in any of the three benchmarks, while our TS-LLaVA outperforms VideoTree on NExT-QA and IntentQA. Unlike VideoTree, which utilizes GPT-4, our model leverages an LLM with only 34B parameters. This shows that our Thumbnail-and-Sampling strategy effectively organizes visual tokens to enhance image LLM's video understanding capabilities.

Following IG-VLM (Kim et al., 2024), we also report results on the sub-tasks of NExT-QA in Table 4. Using only the grid-view image, IG-VLM performs reasonably well on these tasks. However, our TS-LLaVA shows better understanding of the video contents across all aspects, especially in causal and temporal tasks that require more sophisticated temporal

Table 4: Performance (Accuracy) on NExT-QA sub-tasks. We use the same image LLM backbone (LLaVA-v1.6) as IG-VLM.

| Method | NExT-QA sub-tasks | | | |
| | Casual | Temporal | Descriptive | Average |
|---|---|---|---|---|
| IG-VLM (Kim et al., 2024) (7B) | 63.1 | 57.3 | 74.9 | 63.1 |
| TS-LLaVA (Ours, 7B) | **66.4** | **62.0** | **75.8** | **66.5** |
| IG-VLM (Kim et al., 2024) (34B) | 72.2 | 65.7 | 77.3 | 70.9 |
| TS-LLaVA (Ours, 34B) | **74.6** | **68.2** | **81.5** | **73.6** |

Table 3: Overall performance (Accuracy) obtained on *Multiple Choice VideoQA*. We **highlight** the top-performing training-free methods and underline the best-performing video LLMs overall. Methods below the dashed line (- -) are training-free, while those above it have been trained on extensive video data.

| Method | LLM Size | Vision Encoder | NExT-QA | EgoSchema | IntentQA |
|---|---|---|---|---|---|
| Video-LLaVA (Lin et al., 2023) | 7B | ViT-L | 60.5 | 37.0 | - |
| Video-LLaMA2 (Cheng et al., 2024) | 7B | CLIP-L | - | 51.7 | - |
| MovieChat+ (Song et al., 2024b) | 7B | CLIP-G | 54.8 | 56.4 | - |
| Vista-LLaMA (Ma et al., 2024) | 7B | CLIP-G | 60.7 | - | - |
| DeepStack-L (Meng et al., 2024) | 7B | CLIP-L | 61.0 | 38.4 | - |
| $M^3$ (Cai et al., 2024) | 7B | CLIP-L | 63.1 | 36.8 | 58.8 |
| IG-VLM (Kim et al., 2024) | 7B | CLIP-L | 63.1 | 35.8 | 60.3 |
| SF-LLaVA (Xu et al., 2024b) | 7B | CLIP-L | 64.2 | 47.2 | 60.1 |
| TS-LLaVA (Ours) | 7B | CLIP-L | **66.5** | **50.2** | **61.7** |

(a) All models use 7B or comparable LLMs.

| Method | LLM Size | Vision Encoder | NExT-QA | EgoSchema | IntentQA |
|---|---|---|---|---|---|
| Video-LLAMA2 (Cheng et al., 2024) | 46.7B | CLIP-L | - | 53.3 | - |
| LLoVi (Zhang et al., 2024a) | GPT-3.5 | Unknown | 67.7 | 50.3 | 64.0 |
| VideoAgent (Wang et al., 2024a) | GPT-4 | Unknown | 71.3 | 60.2 | - |
| VideoTree (Wang et al., 2024b) | GPT-4 | Unknown | 73.5 | **66.2** | 66.9 |
| IG-VLM (Kim et al., 2024) | 34B | CLIP-L | 70.9 | 53.6 | 65.3 |
| SF-LLAVA (Xu et al., 2024b) | 34B | CLIP-L | 72.0 | 55.8 | 66.5 |
| TS-LLaVA (Ours) | 34B | CLIP-L | **73.6** | 57.8 | **67.9** |

(b) All models use 34B or stronger LLMs.

reasoning. Complementing thumbnail image tokens with sampled tokens provides the image LLM with more effective visual cues.

### 4.2.2 Multitask Benchmarks

To comprehensively evaluate our TS-LLaVA, we conduct experiments on two challenging multitask video understanding benchmarks: MVBench and MLVU. Aiming at testing the limit of our method, we mainly compare our TS-LLaVA to training-based methods on these two challenging benchmarks.

**MVBench** We present the results in Table 5a. Among the training-free methods, our 34B model outperforms the proprietary model GPT-4V by a large margin, in both average accuracy and across most sub-tasks. Even our 7B model surpasses GPT-4V in average accuracy, showing the strong understanding capability and potential of our method.

When comparing to the training-based video LLM, we focus on PLLaVA (Xu et al., 2024a). PLLaVA uses the same image LLM backbone as our model but is further trained on video data. In over half of the sub-tasks, our TS-LLaVA manages to obtain comparable or better performance than PLLaVA. However, there are still tasks where performance can be improved: (1) On some action centric tasks, TS-LLaVA delivers satisfactory results (AC and AP). However, it struggles with other action-centric tasks (e.g., AA, AL, and AS). (2) TS-LLaVA performs less effectively on tasks that require reasoning over moving objects (MA, MC and MD). (3) TS-LLaVA also shows lower performance on CI and OE tasks.

Given the nature of these task types, TS-LLaVA's lower performance is unsurprising. All of these tasks involve data types not seen during image LLM training, such as actions and moving objects. Without

---

[3]AA: Action Antonym; AC: Action Count; AL: Action Localization; AP: Action Prediction; AS: Action Sequence; CO: Character Order; CI: Counterfactual Inference; EN: Egocentric Navigation; ER: Episodic Reasoning; FA: Fine-grained Action; FP: Fine-grained Pose; MA: Moving Attribute; MC: Moving Count; MD: Moving Direction; OE: Object Existence; OI: Object Interaction; OS: Object Shuffle; ST: Scene Transition; SC: State Change; UA: Unexpected Action.

Table 5: Results obtained on Multitask Benchmarks. We **highlight** the top-performing training-free methods and underline the best-performing video LLMs overall. Methods below the dashed line (- -) are training-free, while those above it have been trained on extensive video data. We denote performance better than or comparable to (■) or lags behind (■) the main competing training-based video LLM (on MVBench: PLLaVA-34B, which uses the same backbones as TS-LLaVA; on MLVU: Video-LLaMA2-72B).

| Method | LLM Size | Vision Encoder | AA | AC | AL | AP | AS | CO | CI | EN | ER | FA | FP | MA | MC | MD | OE | OI | OS | ST | SC | UA | Avg. |
|---|---|---|---|---|---|---|---|---|---|---|---|---|---|---|---|---|---|---|---|---|---|---|---|
| Video-ChatGPT (Maaz et al., 2024) | 7B | CLIP-L | 62.0 | 30.5 | 20.0 | 26.0 | 23.5 | 33.0 | 35.5 | 29.5 | 26.0 | 22.5 | 29.0 | 39.5 | 25.5 | 23.0 | 54.0 | 28.0 | 40.0 | 31.0 | 48.5 | 26.5 | 32.7 |
| Video-LLaMA (Zhang et al., 2023) | 7B | CLIP-G | 51.0 | 34.0 | 22.5 | 25.5 | 27.5 | 40.0 | 37.0 | 30.0 | 21.0 | 29.0 | 32.5 | 32.5 | 22.5 | 22.5 | 48.0 | 40.5 | 38.0 | 43.0 | 45.5 | 39.0 | 34.1 |
| VideoChat (Li et al., 2024a) | 7B | CLIP-G | 56.0 | 35.0 | 27.0 | 26.5 | 33.5 | 41.0 | 36.0 | 23.5 | 23.5 | 33.5 | 26.5 | 42.5 | 20.5 | 25.5 | 53.0 | 40.5 | 30.0 | 48.5 | 46.0 | 40.5 | 35.5 |
| VideoChat2 (Li et al., 2024b)† | 7B | UMT-L | 83.5 | 39.0 | 23.0 | 47.5 | 66.0 | 36.5 | 65.5 | 35.0 | 40.5 | 49.5 | 49.0 | 58.5 | 42.0 | 23.0 | 58.0 | 71.5 | 42.5 | 88.5 | 44.0 | 60.0 | 51.1 |
| VideoChat2 (Li et al., 2024b)‡ | 7B | UMT-L | 83.5 | 37.0 | 44.0 | 58.0 | 75.5 | 47.0 | 72.5 | 35.0 | 37.0 | 50.5 | 66.5 | 87.5 | 64.5 | 47.5 | 87.5 | 74.5 | 45.0 | 82.5 | 51.0 | 60.4 |
| ST-LLM (Liu et al., 2025)* | 7B | ViT-G | 84.0 | 36.5 | 31.0 | 53.5 | 66.0 | 46.5 | 58.5 | 34.5 | 41.5 | 44.0 | 44.5 | 78.5 | 56.5 | 42.5 | 80.5 | 73.5 | 38.5 | 86.5 | 43.0 | 58.5 | 54.9 |
| PLLaVA (Xu et al., 2024a)* | 7B | CLIP-L | 55.5 | 35.0 | 26.0 | 49.0 | 58.0 | 53.5 | 31.0 | 30.5 | 48.0 | 41.0 | 42.0 | 52.0 | 42.0 | 23.5 | 56.0 | 61.0 | 36.0 | 82.0 | 45.0 | 61.0 | 46.6 |
| PLLaVA (Xu et al., 2024a)* | 34B | CLIP-L | 82.0 | 40.5 | 49.5 | 53.0 | 67.5 | 66.5 | 59.0 | 39.5 | 63.5 | 47.0 | 50.0 | 70.0 | 43.0 | 37.5 | 68.5 | 67.5 | 36.5 | 91.0 | 51.5 | 79.0 | 58.1 |
| GPT-4V | GPT4 | Unknown | 72.0 | 39.0 | 40.5 | 63.5 | 55.5 | 52.0 | 11.0 | 31.0 | 59.0 | 46.5 | 47.5 | 22.5 | 12.0 | 12.0 | 18.5 | 59.0 | 29.5 | 83.5 | 45.0 | 73.5 | 43.5 |
| TS-LLaVA (Ours) | 7B | CLIP-L | 58.5 | 41.0 | 27.0 | 53.5 | 54.0 | 53.0 | 32.0 | 32.5 | 45.0 | 36.0 | 38.0 | 49.0 | **30.0** | 23.5 | **58.5** | 59.5 | 32.0 | 85.0 | 43.5 | 58.0 | 45.5 |
| TS-LLaVA (Ours) | 34B | CLIP-L | **73.5** | **46.5** | 39.0 | 47.5 | **61.0** | **66.5** | 39.0 | **42.5** | **65.0** | 46.5 | 47.5 | **51.5** | **30.0** | 24.0 | 52.5 | 65.5 | 37.5 | 89.5 | 45.5 | **80.5** | 52.6 |

(a) MVBench results[3]. †: Vicuna as LLM; ‡: Mistral as LLM. The other 7B/34B models use Vicuna as LLM. *: results reported in (Xu et al., 2024a), the rest are from the official repository/paper (Li et al., 2024b).

| Method | LLM Size | Vision Encoder | Holistic LVU | | | Single-Detail LVU | | | Multi-Detail LVU | | | Avg. |
|---|---|---|---|---|---|---|---|---|---|---|---|---|
| | | | Topic Rea. | Anomaly Rec. | Needle QA | Ego Rea. | Plot QA | Sports QA | Act. Order | Act. Count | Tutorial QA | |
| VideoChat (Li et al., 2024a) | 7B | CLIP-G | 26.4 | 12.8 | 18.3 | 17.0 | 22.0 | 11.1 | 15.7 | 11.7 | 14.0 | 16.6 |
| Video-ChatGPT (Maaz et al., 2024) | 7B | CLIP-L | 17.6 | 17.9 | 28.3 | 32.1 | 22.0 | 27.8 | 17.1 | 13.3 | 11.6 | 20.9 |
| Video-LLaMA2 (Cheng et al., 2024) | 13B | CLIP-L | 52.7 | 12.8 | 13.3 | 17.0 | 12.0 | 19.4 | 15.7 | 8.3 | 18.6 | 18.9 |
| VideoChat2 (Li et al., 2024b) | 7B | UMT-L | 72.5 | 30.8 | 18.3 | 28.3 | 26.0 | 36.1 | 17.1 | 23.3 | 18.6 | 30.1 |
| Video-LLaVA (Lin et al., 2023) | 7B | ViT-L | 70.3 | 38.5 | 13.3 | 26.4 | 26.0 | 38.9 | 20.0 | 21.7 | 20.9 | 30.7 |
| LLaMA-VID (Li et al., 2024c) | 7B | CLIP-G | 20.9 | 23.1 | 21.7 | 11.3 | 16.0 | 16.7 | 18.6 | 15.0 | 11.6 | 17.2 |
| MovieChat (Song et al., 2024a) | 7B | CLIP-G | 18.7 | 10.3 | 23.3 | 15.1 | 16.0 | 30.6 | 17.1 | 15.0 | 16.3 | 18.0 |
| Video-LLaMA2 (Cheng et al., 2024) | 13B | CLIP-L | 52.7 | 12.8 | 13.3 | 17.0 | 12.0 | 19.4 | 15.7 | 8.3 | 18.6 | 18.9 |
| Video-LLaMA2 (Cheng et al., 2024) | 72B | CLIP-L | 80.2 | 53.8 | 36.7 | 54.7 | 54.0 | 38.9 | 42.9 | 16.7 | 32.6 | 45.6 |
| LLaVA-NeXT-Image (Zhang et al., 2024d) | 7B | CLIP-L | 63.7 | 17.9 | 13.3 | 26.4 | 30.0 | 22.2 | 21.4 | 16.7 | 16.3 | 25.3 |
| TS-LLaVA (Ours) | 7B | CLIP-L | 76.9 | 38.5 | 28.3 | 34.0 | 30.0 | 30.6 | 25.7 | **20.0** | 18.6 | 33.6 |
| TS-LLaVA (Ours) | 34B | CLIP-L | **83.5** | **43.6** | **55.0** | 32.1 | **46.0** | **55.6** | **28.6** | 10.0 | **32.6** | **43.0** |

(b) Performance on multiple choice questions in MLVU-Test. Rea.: Reasoning; Rec.: Recognition; QA: Question Answering; Act.: Action.

specific training on these data types, bridging the gap between our training-free approach and training-based methods remains challenging.

**MLVU** The results are presented in Table 5b. Our method significantly outperforms the training-free counterpart, establishing the SOTA results across all sub-tasks.

In comparison with training-based video LLM, we take a challenging opponent, namely a 72B Video-LLaMA2. Remarkably, TS-LLaVA-34B achieves comparable or even superior performance to Video-LLaMA2-72B on more than half of the sub-tasks, despite using a much smaller LLM.

For tasks where TS-LLaVA lags behind, we observe a pattern similar to MVBench. Tasks involving less familiar data types for image LLMs, such as action order, pose greater challenges for our training-free approach.

**Overall** Our TS-LLaVA shows promising results on two challenging multitask benchmarks, but inherent limitations of image LLMs make some tasks difficult. Despite being a training-free method, our TS-LLaVA significantly outperforms several training-based approaches on certain tasks. This demonstrates that, by thoroughly investigating effective compression strategies, we can build video LLMs without relying on large-scale video-text paired datasets, such as the 783k pairs used in PLLaVA or the over 10 million pairs required by VideoLLaMA2.

### 4.2.3 Strong Potential in Long Video Understanding

On MLVU-test, TS-LLaVA has demonstrated exceptional performance in long video understanding tasks, even when compared to training-based video LLMs. In this section, we present additional results on long video understanding benchmarks, comparing TS-LLaVA with previous training-free video LLMs, specifically IG-VLM and SF-LLaVA.

As shown in Table 6, thanks to the advanced compression strategy, our TS-LLaVA achieves superior performance across all three long video understanding benchmarks compared to previous training-free video

Table 6: Accuracy on EgoSchema, VideoMME and LongVideoBench. The same image LLM backbone is adopted for three models, namely LLaVA-v1.6.

| Model | Size | EgoSchema | VideoMME | LongVideoBench | Size | EgoSchema | VideoMME | LongVideoBench |
|-------|------|-----------|----------|----------------|------|-----------|----------|----------------|
| IG-VLM (Kim et al., 2024) | 7B | 35.8 | 39.8 | 38.1 | 34B | 53.6 | 50.9 | 49.8 |
| SF-LLaVA (Xu et al., 2024b) | 7B | 47.2 | 41.2 | 41.3 | 34B | 55.8 | 52.9 | 53.9 |
| TS-LLaVA (Ours) | 7B | **50.2** | **44.8** | **43.1** | 34B | **57.8** | **55.1** | **56.2** |

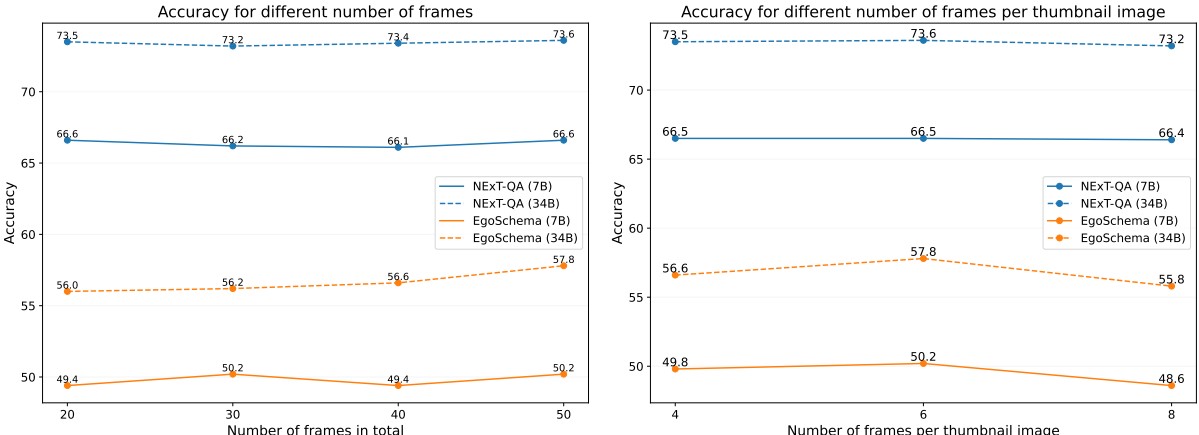

(a) Performance of TS-LLaVA with different number of frames in total. The total number of visual tokens is the same (3456).

(b) Performance of TS-LLaVA with different number of frames per thumbnail image. We use 50 frames with 3456 tokens in total.

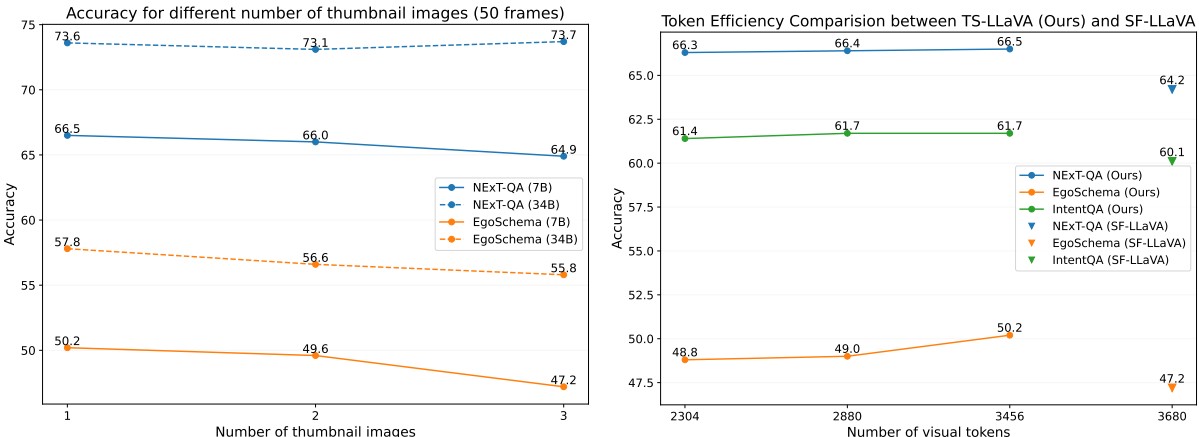

(c) Performance of TS-LLaVA with different number of thumbnail images used. The total number of visual tokens is the same (3456).

(d) Token efficiency comparison between the previous SOTA training-free video LLM SF-LLaVA. We compare 7B models here.

Figure 4: Design choices of TS-LLaVA. In (a), (b) (c), IntentQA shows similar pattern as NExT-QA, please refer to the Appendix.

LLMs. Notably, despite using fewer visual tokens than SF-LLaVA (3,456 vs. 3,680), TS-LLaVA still outperforms SF-LLaVA by a significant margin. This highlights the effectiveness of our compression strategy in efficiently compressing visual information from long video sequences. We further discuss the comparison with SF-LLaVA in Section 4.4.

### 4.3 Design Choices of TS-LLaVA

We dive deeper into the design choices of TS-LLaVA in this section. More results can be found in Section B.3 in the Appendix

**Number of frames** We vary the maximum number of input frames used in TS-LLaVA, and present the results in Figure 4a. Since the total number of visual tokens remains constant at 3456, using fewer frames results in a lower compression rate. For datasets like NExT-QA, which do not emphasize long-term video understanding, the reduced compression rate effectively compensates for any missing information from fewer frames. In contrast, for EgoSchema, which specifically targets long-term understanding, using more frames proves beneficial. One interesting observation is that the 7B and 34B models behave differently to reduced frame counts. The performance of the 34B model keeps increasing as we increase the number of frames, indicating it can handle more information as we increase frame numbers.

**How many frames per thumbnail image?** We conduct experiments with varying numbers of frames per thumbnail image. The results are presented in Figure 4b. Since the resolution of each image is fixed at 336×336 for the image encoder, including more frames in the thumbnail image means lower resolution for each frame. The results show that changing the number of frames does not affect the performance on NExT-QA and IntentQA significantly. While for EgoSchema, which requires better temporal understanding and involves longer videos, is more sensitive to the number of frames per thumbnail image. Using 6 frames per thumbnail image shows clear benefit over the counterparts, by providing enough temporal cues and not losing too much details due to reduced resolution.

**How many thumbnail images?** To study the impact of the number of thumbnail images on performance, we report results for 1, 2, and 3 thumbnails in Figure 4c. Each thumbnail corresponds to 576 visual tokens, and increasing the number of thumbnails raises the compression rate for sampled visual tokens. With 50 input frames, the compression rate is $\frac{50\times576}{3456-576\times k}$, where $k$ is the number of thumbnails. Higher compression significantly degrades performance on EgoSchema, which cannot be offset by additional thumbnail tokens. This suggests that tasks requiring temporal understanding benefit from less compressed visual tokens.

**Token efficiency** Image LLMs are not trained on videos involving multiple frames, making token efficiency a critical factor for training-free video LLMs. Compared to the previous SOTA training-free video LLM SF-LLaVA, we gradually decrease the number of visual tokens in TS-LLaVA, and present the results in Figure 4d. Our TS-LLaVA consistently outperforms SF-LLaVA, even with roughly 60% of the number of visual tokens used by SF-LLaVA (2304 vs 3680). Our Thumbnail-and-Sampling strategy can better compress the visual tokens than the slow-fast operation used by SF-LLaVA. We extend the discussion in Section 4.4.

### 4.4 Discussion

**Contributions** We propose a simple yet effective compression strategy for video LLMs, achieving strong performance across multiple benchmarks. This shows that high performance can be achieved without training, challenging the necessity of compute-intensive training-based methods and setting a strong baseline for future research.

Designing proper compression strategy for video LLMs is a challenging and non-trivial task. i) IG-VLM's *Grid* layout struggles with long-term relationships due to limited frames. We extend it to *Grids*, which still hampers fine-grained action recognition from reduced resolution. In contrast, TS-LLaVA, leveraging sampled tokens, handles these tasks far more effectively. ii) Our Thumbnail-and-Sampling strategy is similar to the widely used slow-fast operations in video understanding (Feichtenhofer et al., 2019; Xiao et al., 2020; Huang et al., 2024; Xu et al., 2024b), including the previous SOTA training-free video LLM, SF-LLaVA. Both use a two-stream design for processing video frames, but the key difference lies in how frames are handled.

- SF-LLaVA : The slow pathway retains fewer frames at a lower frame rate but with a lower compression rate. The fast pathway includes more frames at a higher frame rate but applies a higher compression rate.

- TS-LLaVA : The Thumbnail pathway uses fewer frames at a lower resolution. The Sampling pathway incorporates more frames at a higher resolution, retaining only sampled token information.

In SF-LLaVA, most of the token budget (2880 out of 3680 tokens) is allocated to the slow pathway, resulting in a high compression rate of 36 in the fast pathway with 50 frames. In contrast, our method preserves more information in the Sampling pathway by allocating 2880 tokens to 50 frames, achieving a lower compression rate of 10. This allows us to capture long-term dependencies more effectively while maintaining better token efficiency, as shown in Figure 4d. Consequently, our method enhances token efficiency in representing video content, offering a significant advantage.

**Limitations** Our focus is on visual token compression rather than prompt design. While prior works highlight prompt effects (Kim et al., 2024), our preliminary experiments show limited impact. Reduced resolution from Thumbnail images can be addressed by high-resolution operations in image LLMs (Liu et al., 2024a), which use image patch features. Combining vision encoders via feature routing (Qu et al., 2025) also holds promise. We leave these explorations for future work.

## 5 Conclusion

In this work, we thoroughly evaluate the advantages and limitations of various visual token compression strategies for training-free video LLMs. Our findings lead to TS-LLaVA, a training-free video LLM leveraging the novel Thumbnail-and-Sampling compression strategy. This approach provides both a summarized view of the video (Thumbnail) and a detailed representation of long-term temporal relations (Sampling). As a simple yet strong baseline, TS-LLaVA achieves state-of-the-art performance among training-free video LLMs and outperforms several competitive training-based models. Our extensive experiments provide valuable insights for future video LLMs.

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

# A   Additional Implementation Details

To construct the thumbnail image, we arrange the selected frames in a 2-column by 3-row grid. Each frame is resized accordingly to fit within the resulting 336×336 thumbnail image.

**MVBench Details** We report the detailed classification of each sub-task included in MVBench:

- *Action*: Action Sequence (AS), Action Antonym (AA), Action Prediction (AP), Unexpected Action (UA) and Fine-grained Action (FA)

- Object: Object Shuffle (OS), Object Existence (OE) and Object Interaction (OI)

- *Position*: Moving Direction (MD) and Action Localization (AL)

- *Count*: Action Count (AC) and Moving Count (MC)

- *Scene*: Scene Transition (ST)

- *Pose*: Fine-grained Pose (FP)

- *Attribute*: State Change (SC) and Moving Attribute (MA)

- *Character*: Character Order (CO)

- *Cognition*: Episodic Reasoning (ER), Egocentric Navigation (EN) and Counterfactual Inference (CI)

# B   Additional Results

We report additional experimental results in this section. We start with additional experiments conducted for the study on compression strategies. Then we report the results on Open-Ended VideoQA and video-based Text Generation. Finally, we conclude this section with more results on design choices of TS-LLaVA.

## B.1 Compression Strategies

**Max. pooling or Average Pooling** Here we present comparison between performance of LLaVA-v1.6-7B equipped with max. pooling and average pooling on Video-MME in Table 7. For the most cases, average pooling shows superior performance than max. pooling.

Table 7: Comparison between different pooling schemes on Video-MME dataset using LLaVA-v1.6-7B."Comp. Rate": Compression Rate. Here we use a $2 \times 2$ kernel with stride=2 and a $3 \times 3$ kernel with stride=3 for compression rate=2 and 3, respectively.

| Pooling Scheme | #. of Frames ($N$) | #. of Vis. Tokens | Comp. Rate | Video-MME (Accuracy) | | | |
|---|---|---|---|---|---|---|---|
| | | | | Short | Medium | Long | Overall |
| *Max.* | 4 | 768 | 3 | 44.6 | 37.9 | 34.0 | 38.8 |
| *Average* | 4 | 768 | 3 | 46.4 | 40.8 | 36.4 | 41.2 |
| *Max.* | 4 | 1152 | 2 | 46.4 | 39.7 | 36.2 | 40.8 |
| *Average* | 4 | 1152 | 2 | 47.9 | 41.2 | 36.3 | 41.8 |
| *Max.* | 8 | 1536 | 3 | 47.3 | 39.8 | 35.6 | 40.9 |
| *Average* | 8 | 1536 | 3 | 48.6 | 42.9 | 35.8 | 42.4 |
| *Max.* | 8 | 2304 | 2 | 49.1 | 41.7 | 35.6 | 42.1 |
| *Average* | 8 | 2304 | 2 | 48.7 | 41.9 | 36.4 | 42.3 |

## B.2 Open-Ended VideoQA and Text Generation

In this section, we first report the results on Open-Ended VideoQA and Text Generation tasks. Then we point out the problem of existing evaluation protocol.

**Open-Ended VideoQA benchmarks** We evaluate our method on MSVD-QA (Chen & Dolan, 2011), MSRVTT-QA (Xu et al., 2016), TGIF-QA (Li et al., 2016) and ActivityNet-QA (Yu et al., 2019).

**Text Generation benchmark** VCGBench (Maaz et al., 2024) in terms of Correctness of Information (CI), Detail Orientation (DO), Contextual Understanding (CU), Temporal Understanding (TU), and Consistency (CO).

**Evaluation protocol** Following the common practice, we use GPT-assisted evaluation for Open-Ended VideoQA and Text Generation benchmarks. The evaluation assesses the accuracy of the response (in terms of true or false) and the quality of the generated text (score ranging from 0 to 5). As pointed by Wu (2024); Xu et al. (2024b), different GPT versions have significant impacts on the evaluation results. To keep a fair comparison, we adopt GPT-3.5-Turbo-0125 as in (Wu, 2024; Xu et al., 2024b).

**Performance** We report the results on the two tasks in Table 8. Our method outperforms the previous SOTA SF-LLaVA on majority of the tasks evaluated. We consider the evaluation protocol for Open-Ended VideoQA and video-based Text Generation not as reliable as Multiple Choice VideoQA. Hence, we do not discuss too much on the results. Next, we provide case studies on the evaluation protocol.

**Unreliable evaluation with GPT** Both Open-Ended VideoQA and Text Generation tasks require GPT-assisted evaluation. We take Open-Ended VideoQA's evaluation pipeline to showcase the problem. We follow the standard evaluation protocol as adopted by Xu et al. (2024b). The standard prompt for Open-Ended VideoQA evaluation has two parts as:

- "role": "system",

  "content": "You are an intelligent chatbot designed for evaluating the correctness of generative outputs for question-answer pairs. " "Your task is to compare the predicted answer with the correct answer and determine if they match meaningfully. Here's how you can accomplish the task:" "——" "##INSTRUCTIONS: " "- Focus on the meaningful match between the predicted answer and the correct answer.\n" "- Consider synonyms or paraphrases as valid matches.\n" "- Evaluate the correctness of the prediction compared to the answer."

Table 8: *Open-Ended VideoQA* and *Text Generation* results. We **highlight** the top-performing training-free methods and underline the best-performing video LLMs overall. Methods below the dashed line (- -) are training-free, while those above it have been trained on extensive video data. ▇: performance better than SF-LLaVA; ▇: performance comparable to SF-LLaVA

| Method | LLM Size | Vision Encoder | Open-Ended VideoQA (Accuracy/Score) | | | | Text Generation (Score) | | | | | |
|---|---|---|---|---|---|---|---|---|---|---|---|---|
| | | | MSVD-QA | MSRVTT-QA | TGIF-QA | ANet-QA | CI | DO | CU | TU | CO | Avg. |
| Video-LLaMA (Zhang et al., 2023) | 7B | CLIP-G | 51.6/2.5 | 29.6/1.8 | - | 12.4/1.1 | 1.96 | 2.18 | 2.16 | 1.82 | 1.79 | 1.98 |
| Video-LLaMA2 (Cheng et al., 2024) | 7B | CLIP-L | 70.9/3.8 | - | - | 50.2/3.3 | 3.16 | 3.08 | 3.69 | 2.56 | 3.14 | 3.13 |
| Video-ChatGPT (Maaz et al., 2024) | 7B | CLIP-L | 64.9/3.3 | 49.3/2.8 | 51.4/3.0 | 35.2/2.7 | 2.50 | 2.57 | 2.69 | 2.16 | 2.20 | 2.42 |
| VideoGPT+ (Maaz et al., 2024) | 3.8B | CLIP-L | 72.4/3.9 | 60.6/3.6 | 74.6/4.1 | 50.6/3.6 | 3.27 | 3.18 | 3.74 | 2.83 | 3.39 | 3.28 |
| Video-LLaVA (Lin et al., 2023) | 7B | ViT-L | 70.7/3.9 | 59.2/3.5 | 70.0/4.0 | 45.3/3.3 | - | - | - | - | - | - |
| MovieChat (Song et al., 2024a) | 7B | CLIP-G | 75.2/3.8 | 52.7/2.6 | - | 45.7/3.4 | 2.76 | 2.93 | 3.01 | 2.24 | 2.42 | 2.67 |
| MovieChat+ (Song et al., 2024b) | 7B | CLIP-G | 76.5/3.9 | 53.9/2.7 | - | 48.1/3.4 | - | - | - | - | - | - |
| VideoChat (Li et al., 2024a) | 7B | CLIP-G | 56.3/2.8 | 45.0/2.5 | 34.4/2.3 | 26.5/2.2 | 2.23 | 2.50 | 2.53 | 1.94 | 2.24 | 2.29 |
| VideoChat2 (Li et al., 2024b) | 7B | UMT-L | 70.0/3.9 | 54.1/3.3 | - | 49.1/3.3 | 3.02 | 2.88 | 3.51 | 2.66 | 2.81 | 2.98 |
| Vista-LLaMA (Ma et al., 2024) | 7B | CLIP-G | 65.3/3.6 | 60.5/3.3 | - | 48.3/3.3 | 2.44 | 2.64 | 3.18 | 2.26 | 2.31 | 2.57 |
| LLaMA-VID (Li et al., 2024c) | 13B | CLIP-G | 69.7/3.7 | 57.7/3.2 | - | 47.4/3.3 | 2.96 | 3.00 | 3.53 | 2.46 | 2.51 | 2.89 |
| PLLaVA (Xu et al., 2024a) | 7B | CLIP-L | 76.6/4.1 | 62.0/3.5 | 77.5/4.1 | 56.3/3.5 | - | - | - | - | - | - |
| LLaVA-NeXT-Video (Zhang et al., 2024d) | 7B | CLIP-L | - | - | - | 53.5/3.2 | 3.39 | 3.29 | 3.92 | 2.60 | 3.12 | 3.26 |
| LLaVA-NeXT-Video-DPO (Zhang et al., 2024d) | 7B | CLIP-L | - | - | - | 60.2/3.5 | 3.64 | 3.45 | 4.17 | 2.95 | 4.08 | 3.66 |
| FreeVA (Wu, 2024) | 7B | CLIP-L | 73.8/4.1 | 60.0/3.5 | - | 51.2/3.5 | - | - | - | - | - | - |
| DeepStack-L (Meng et al., 2024) | 7B | CLIP-L | 76.0/4.0 | - | - | 49.3/3.1 | - | - | - | - | - | - |
| LLaVA-NeXT-Image (Zhang et al., 2024d) | 7B | CLIP-L | - | - | - | 53.8/3.2 | 3.05 | **3.12** | **3.68** | 2.37 | 3.16 | 3.07 |
| IG-VLM (Kim et al., 2024) | 7B | CLIP-L | 78.8/4.1 | 63.7/3.5 | 73.0/4.0 | 54.3/3.4 | 3.11 | 2.78 | 3.51 | 2.44 | 3.29 | 3.03 |
| SF-LLaVA (Xu et al., 2024b) | 7B | CLIP-L | 79.1/4.1 | 65.8/3.6 | 78.7/4.2 | 55.5/3.4 | 3.09 | 2.70 | 3.57 | 2.52 | **3.35** | 3.04 |
| TS-LLaVA (Ours) | 7B | CLIP-L | 79.0/4.1 | 65.1/3.6 | 77.7/4.1 | **56.7/3.4** | **3.18** | 2.82 | 3.58 | **2.53** | 3.34 | **3.09** |

(a) All models use 7B or comparable LLMs.

| Method | LLM Size | Vision Encoder | Open-Ended VideoQA (Accuracy/Score) | | | | Text Generation (Score) | | | | | |
|---|---|---|---|---|---|---|---|---|---|---|---|---|
| | | | MSVD-QA | MSRVTT-QA | TGIF-QA | ANet-QA | CI | DO | CU | TU | CO | Avg. |
| Video-LLAMA2 (Cheng et al., 2024) | 46.7B | CLIP-L | 70.5/3.8 | - | 50.3/3.4 | - | 3.08 | 3.11 | 3.64 | 2.67 | 3.26 | 3.15 |
| PLLaVA (Xu et al., 2024a) | 34B | CLIP-L | 79.9/4.2 | 68.7/3.8 | 80.6/4.3 | 60.9/3.7 | - | - | - | - | - | - |
| LLaVA-NeXT-Video (Zhang et al., 2024d) | 34B | CLIP-L | - | - | - | 58.8/3.4 | 3.48 | 3.37 | 3.95 | 2.64 | 3.28 | 3.34 |
| LLaVA-NeXT-Video-DPO (Zhang et al., 2024d) | 34B | CLIP-L | - | - | - | 64.4/3.6 | 3.81 | 3.55 | 4.24 | 3.14 | 4.12 | 3.77 |
| LLaVA-NeXT-Image (Zhang et al., 2024d) | 34B | CLIP-L | - | - | - | 55.6/3.3 | 3.29 | **3.23** | 3.83 | 2.51 | 3.47 | 3.27 |
| IG-VLM (Kim et al., 2024) | 34B | CLIP-L | 79.6/4.1 | 62.4/3.5 | 79.1/4.2 | 58.4/3.5 | 3.21 | 2.87 | 3.54 | 2.51 | 3.34 | 3.09 |
| SF-LLAVA (Xu et al., 2024b) | 34B | CLIP-L | 79.9/4.1 | 67.4/3.7 | 80.6/4.3 | 59.2/3.5 | 3.48 | 2.96 | 3.84 | 2.77 | 3.57 | 3.32 |
| TS-LLaVA (Ours) | 34B | CLIP-L | 79.4/4.1 | 66.2/3.6 | **81.0/**4.2 | 58.9/3.5 | **3.55** | 3.03 | **3.86** | 2.77 | **3.69** | **3.38** |

(b) All models use 34B or stronger LLMs.

- "role": "user",

  "content": "Please evaluate the following video-based question-answer pair:\n\n" "Question: *Question*\n" "Correct Answer: *Answer*\n" "Predicted Answer: *Prediction*\n\n" "Provide your evaluation only as a yes/no and score where the score is an integer value between 0 and 5, with 5 indicating the highest meaningful match. " "Please generate the response in the form of a Python dictionary string with keys 'pred' and 'score', where value of 'pred' is a string of 'yes' or 'no' and value of 'score' is in INTEGER, not STRING." "DO NOT PROVIDE ANY OTHER OUTPUT TEXT OR EXPLANATION. Only provide the Python dictionary string. " "For example, your response should look like this: 'pred': 'yes', 'score': 4.8."

where *Question*, *Answer* and *Prediction* are the place holder for question, ground truth answer and model prediction, respectively.

We present two randomly selected cases from the Open-Ended VideoQA evaluation pipeline. For each case, we run the evaluation code for 50 times. The recorded evaluation result is also presented as *Recorded Prediction*: {"pred": , "score":}

**Case 1 (from TGIF-QA)**:

*Question*: what is the very fit male?

*Answer*: underwear

*Prediction*: there is a sequence of frames showing a very fit male. The sequence includes the man standing in various poses, showcasing his muscular physique. The man is shirtless, wearing boxer shorts, and has tattoos on his torso. The sequence captures the man's fitness and athleticism.

*Recorded Prediction*: {"pred": yes, "score":5}

**Case 2 (from MSVD-QA)**:

*Question*: what is getting kicked into the goal?

*Answer*: ball

*Prediction*: there is a sequence of key frames showing a soccer match. The goal is being kicked into it by a player wearing a blue jersey. The sequence captures the progression of the kick, with the ball moving closer to the goal with each frame. The goalkeeper is positioned to react to the shot, and the other players are in various defensive positions. The sequence highlights the action of the player kicking the ball towards the goal, which is a significant event in the context of the soccer match.

*Recorded Prediction*: {"pred": yes, "score":3}

**Results discussion**

The results of two case studies are presented in Table 9. For Case 1, the recorded response is "yes/5", which can be interpreted as clearly correct. While as shown in the table, less than half of the runs (24/50) return the same response. There are even 20% of the runs (10/50) result in "no/2". For Case 2, the recorded response is "yes/3", which shows that the model is not very sure about the assessment. When we run the evaluation for 50 times, the majority of the runs return "no/2".

Table 9: Prediction statistics of the case study for Open-Ended VideoQA evaluation. "Prediction Count" records the number of "yes" (correct prediction) and "no" (incorrect prediction) among the 50 runs. We also present the average evaluation score, and how many times each score appear. The colored text marks the recorded evaluation.

| Case | Prediction Count | | Average | Score Appearance | | | | |
|---|---|---|---|---|---|---|---|---|
| | Yes | No | Score | 1 | 2 | 3 | 4 | 5 |
| 1 | 38 | 12 | 4.02 | 0 | 10 | 3 | 13 | 24 |
| 2 | 8 | 42 | 2.18 | 2 | 40 | 5 | 3 | 0 |

The case study shows that the random-
ness in GPT assisted evaluation is not as reliable as we hope for. We urge for alternative evaluation methods
for Open-Ended VideoQA.

### B.3 Design Choices

In this section, we report the additional results supporting the design choices of our TS-LLaVA, as discussed
in Section 4.3. We first report results from changing positioning of visual tokens, then we report full results
of the experiments conducted in Section 4.3.

**Thumbnail first or sampling first?** We study the effect of whether to prepend or append thumbnail
image tokens to sampled visual tokens. We present the result in Figure 5. We do not observe a clear
difference between this two ways of positioning visual tokens. It also indicates that the backbone image
LLM is capable of handling varying visual token patterns, which offers potentials to future research.

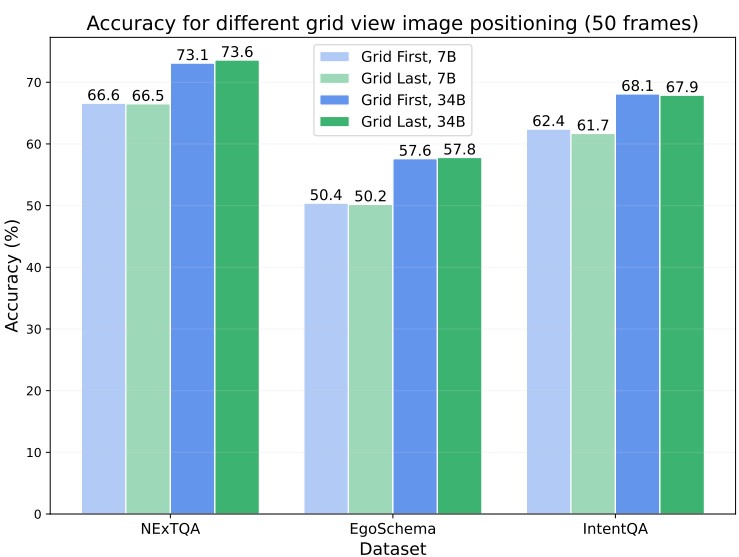

Figure 5: Results from different ways of positioning visual tokens. Grid First: the thumbnail image tokens
are prepended to sampled visual tokens.

**More results on design choices of TS-LLaVA** We report the additional results on design choices of
TS-LLaVA.

Table 10: Performance of TS-LLaVA with different
number of frames in total. The total number of
visual tokens is the same (3456).

| Model Size | #. of Frames | NExT-QA | EgoSchema | IntentQA |
|---|---|---|---|---|
| 7B | 20 | 66.6 | 49.4 | 62.8 |
| 7B | 30 | 66.2 | 50.2 | 62.6 |
| 7B | 40 | 66.1 | 49.4 | 62.3 |
| 7B | 50 | 66.6 | 50.2 | 61.7 |
| 34B | 20 | 73.5 | 56.0 | 68.1 |
| 34B | 30 | 73.2 | 56.2 | 68.0 |
| 34B | 40 | 73.4 | 56.6 | 67.7 |
| 34B | 50 | 73.6 | 57.8 | 67.9 |

Table 11: Performance of TS-LLaVA with different
number of thumbnail images. The total number of
visual tokens is the same (3456).

| Model Size | #. of Thumbnail Img. | NExT-QA | EgoSchema | IntentQA |
|---|---|---|---|---|
| 7B | 1 | 66.5 | 50.2 | 61.7 |
| 7B | 2 | 66.0 | 49.6 | 61.9 |
| 7B | 3 | 64.9 | 47.2 | 62.0 |
| 34B | 1 | 73.6 | 57.8 | 67.9 |
| 34B | 2 | 73.1 | 56.6 | 68.7 |
| 34B | 3 | 73.7 | 55.8 | 68.6 |

The performance of our TS-LLaVA on all three Multiple Choice VideoQA datasets with different number
of input frames can be found in Table 10. It can be seen that for EgoSchema, the impact of varying frame
numbers is bigger than the other two datasets.

We present the results of our TS-LLaVA on all three Multiple Choice VideoQA datasets with varying number of thumbnail images in Table 10. Since the total number of visual tokens is the same, increasing the number of thumbnail images leads to increased compression rate on sampled visual tokens. For EgoSchema, which focuses on the long term temporal dependency in the video, less sampled tokens leads to degraded performance. While for NExT-QA and IntentQA, the details brought by the increased thumbnail images can well mitigate the loss of information from the reduced sampled tokens. Hence, the performance on NExT-QA and IntentQA remains at a relatively high level.

Table 12: Performance of TS-LLaVA on downstream tasks in terms of accuracy varying the number of frames in the thumbnail image. We use the standard setting where only 1 thumbnail image is constructed.

| Model Size | #. of images | NExT-QA | EgoSchema | IntentQA |
|---|---|---|---|---|
| 7B | 4 | 66.5 | 49.8 | 62.0 |
| 7B | 6 | 66.5 | 50.2 | 61.7 |
| 7B | 8 | 66.4 | 48.6 | 61.8 |
| 34B | 4 | 73.5 | 56.6 | 68.0 |
| 34B | 6 | 73.6 | 57.8 | 67.9 |
| 34B | 8 | 73.2 | 55.8 | 68.6 |

Table 13: Performance of TS-LLaVA with different number of visual tokens. We use the standard setting with 50 input frames in total and 1 thumbnail image constructed.

| Model Size | #. of visual tokens | NExT-QA | EgoSchema | IntentQA |
|---|---|---|---|---|
| 7B | 2304 | 66.3 | 48.8 | 61.4 |
| 7B | 2880 | 66.4 | 49.0 | 61.7 |
| 7B | 3456 | 66.5 | 50.2 | 61.7 |
| 34B | 2304 | 73.7 | 56.4 | 68.2 |
| 34B | 2880 | 73.2 | 58.0 | 68.7 |
| 34B | 3456 | 73.6 | 57.8 | 67.9 |

We also report the results on different number of frames used for composing the thumbnail image in Table 12. Using 6 images per thumbnail image leads to overall the best performance. This finding also aligns with Kim et al. (2024).

Finally, the results of using different numbers of visual tokens are presented in Table 13. Since the number of visual tokens resulting from the thumbnail image is constant, varying the total number of visual tokens affects the sampling pathway only. The results show that this change affects EgoSchema more, as it relies on long-term temporal understanding.

