# OpenReview forum: "TS-LLaVA: Constructing Visual Tokens through Thumbnail-and-Sampling for Training-Free Video Large Language Models"
_TMLR — Rejected by TMLR_

### Review · Reviewer_6cP7 · 2025-07-26

**Summary Of Contributions:**

Curating high-quality video-text paired data is challenging. This paper explores a method to adapt an image LLM to video understanding tasks without additional training. Specifically, it analyzes existing approaches that compress visual tokens extracted from individual frames for efficient video encoding, and proposes a novel hybrid method called Thumbnail-and-Sampling. In this method, visual tokens are derived from a thumbnail image created by merging a few equidistant frames, and are combined with a subset of visual tokens sampled from the entire video. The resulting video LLM, TS-LLaVA, outperforms prior training-free video LLMs on multiple-choice video QA datasets and multitask video benchmarks, and achieves performance comparable to training-based video LLMs on some tasks.

**Audience:**

Yes

**Claims And Evidence:**

Yes

**Requested Changes:**

Please address the above concerns.

**Strengths And Weaknesses:**

Strengths
- Overall, the paper is easy-to-read and well written. Figure 2 and Figure 3 help understand the core idea of the proposed method, the Thumbnail-and-Sampling strategy.
- The proposed framework, TS-LLaVA, achieves strong performance on various benchmarks including several multiple-choice video QA and general video understanding benchmarks.
- The break-down analysis of model performance on benchmarks, the ablation study to verify model design choices, and the comparison with the similar previous approach, SF-LLaVA, clearly demonstrate the strengths and weaknesses of TS-LLaVA.

Weaknesses
- The core idea of TS-LLaVA is relatively incremental. Even though sampling encoded visual tokens has been less explored, the notion of constructing a grid thumbnail image is directly borrowed from IG-VLM and using two-stream design with different frame rates is similar to SF-LLaVA. Furthermore, the authors should have explored various sampling strategies, not just uniform sampling. For example, to increase diversity, we can compute similarity scores between encoded visual tokens and sample only those that are not too similar to each other. Alternatively, off-the-shelf models could be used to select only the visual tokens that are closely related to the query.
- The low resolution of thumbnail images seems critical. In addition to AnyRes [1] mentioned in the paper, various techniques have been explored to handle high-resolution images in vision-language models, such as Scaling on Scales (S^2) [2], positional embedding interpolation used in Oryx [3], and dynamic resolution used in Qwen2-VL [4]. It seems necessary to explore a method to increase the resolution of thumbnail images.
- The compression strategies analyzed by the authors either simply concatenate RGB pixel values or encoded visual tokens from multiple frames, without modeling temporal dynamics or inter-frame correlations. It would have been better if such a module had been included.
- TS-LLaVA shows weaknesses in tasks that explicitly require temporal understanding, as it adapts an image LLM to video tasks without training. To further analyze this limitation, it would have been helpful to include evaluations on tasks such as spatio-temporal localization.

Minor request
- Please provide the results of IG-VLM and SF-LLaVA on MVBench.

[1] Haotian Liu et al., Improved Baselines with Visual Instruction Tuning, CVPR 2024

[2] Baifeng Shi et al., When Do We Not Need Larger Vision Models?, ECCV 2024

[3] Zuyan Liu et al., Oryx MLLM: On-Demand Spatial-Temporal Understanding at Arbitrary Resolution, ICLR 2025

[4] Peng Wang et al., Qwen2-VL: Enhancing Vision-Language Model's Perception of the World at Any Resolution, arXiv preprint arXiv:2409.12191

---

> ### Author Response · Authors · 2025-08-30
> **Response to Reviewer 6cP7**
>
> Thank you for acknowledging that our paper is **easy-to-read and well written**, and that our **TS-LLaVA achieves strong performance on various benchmarks**.
> We are also encourage to hear that **the break-down analysis of model performance on benchmarks, the ablation study to verify model design choices, and the comparison with the similar previous approach, SF-LLaVA, clearly demonstrate the strengths and weaknesses of TS-LLaVA.**
>
> ---
> > ### Concerns on novelty
>
> We understand your concerns. However, the significance of our research extends beyond merely introducing a ”novel” method, which, we argue, is not the criterion for impact in a TMLR paper. Our contributions are substantial, and the findings are insightful:
> * We propose a **simple yet effective** compression strategy for video LLMs, achieving strong performance across multiple benchmarks. This shows that high performance can be achieved with a training-free approach, **challenging the necessity of compute-intensive training-based methods and setting a strong baseline for future research**.
> * We also offer deep insights for developing video LLMs.
>     * IG-VLM’s Grid layout struggles with long-term relationships due to limited frames. We extend it to Grids, which still hampers fine-grained action recognition from reduced resolution. In contrast, TS-LLaVA, leveraging sampled tokens, handles these tasks far more effectively.
>     * As also noted in Sec. 4.4, SF-LLaVA uses more frames in fast pathway, where similar adjacent frames lead to less distinctive features after aggressive pooling, harming understanding. In contrast, TS-LLaVA samples tokens from higher frame rates and complements them with summarized thumbnail tokens, preserving more video information.
>     * TS-LLaVA is empirically grounded and delivers significantly better performance. **Designing proper compression strategy for video LLMs is a challenging and non-trivial task**.
>
> ---
> > ### Comments on alternative architecture design
>
>
> Thank you for the suggestions regarding alternative sampling strategies and module designs. While such approaches are promising, our goal is to establish a simple yet effective baseline for video LLMs. To this end, we design all components in their simplest and most straightforward form.
> We aim to demonstrate from first principles that our Thumbnail-then-Sampling strategy is effective and well-suited for video understanding. By providing a strong, minimal baseline, we hope to enable future research, such as exploring adaptive sampling, inter-frame modeling, or other extensions, on a solid and reproducible foundation. We believe this approach will benefit the video LLM community by clarifying core design principles and enabling systematic investigation of more advanced techniques.
>
> ---
> > ### Suggestions on resolutions of thumbnail images
>
> Thank you for the constructive suggestion. We believe that higher resolution can indeed lead to significantly better performance. And we expect to explore more in future studies. As for now, we report results on using Qwen2-VL as the backbone, and compare our method to the long-video tuned video LLM LongVU.
>
> **Performance comparison with Video LLMs on VideoMME**
> | Method | Visual Tokens | Short | Medium | Long | Overall |
> |---|---|---|---|---|---|
> | Qwen2-VL | 34560 | 64.89 | 50.56 | 45.89 | 53.78|
> | LongVU | 4320 | 65.00 | 52.44 | 47.11 | 54.85 |
> | TS-LLaVA (Ours) | 4320 | 66.89 | 52.33 | 45.33 | 54.85 |
>
> ---
> > ### Performance on temporal understanding
>
> Among our selected benchmarks, EgoSchema is a particularly challenging dataset that emphasizes long-form temporal understanding. As shown in Table 3 and below, TS-LLaVA achieves the best performance among training-free methods on this benchmark, demonstrating strong temporal reasoning capabilities.
>
> **Performance comparison on EgoSchema**
> | Method | 7B | 34B |
> |---|---|---|
> | IG-VLM | 35.8 | 53.6 |
> | SF-LLaVA | 47.2 | 55.8 |
> | TS-LLaVA (Ours) | 50.2 | 57.8 |
>
> Similarly, the MLVU dataset consists exclusively of long videos (ranging from 3 minutes to 2 hours) and includes nine tasks specifically designed for long-video understanding (LVU). On the MLVU test set, TS-LLaVA achieves phenomenal performance (see Table 4b).
>
> ---
> > ### Performance comparison on MVBench
>
> As requested, we report the performance of IG-VLM and SF-LLaVA on MVBench. Our TS-LLaVA outperforms these two methods again by a large margin.
>
> **Performance comparison on MVBench**
> | Method | 7B | 34B |
> |---|---|---|
> | IG-VLM | 41.5 | 48.1 |
> | SF-LLaVA | 44.1 | 49.8 |
> | TS-LLaVA (Ours) | 45.5 | 52.6 |

---

### Review · Reviewer_oJ2u · 2025-08-07

**Summary Of Contributions:**

This paper proposes TS-LLaVA, a training-free approach to adapt image-only vision-language models (VLMs) for video understanding tasks. The method constructs a video representation by combining thumbnail frames (grid-based composite images of multiple frames) and uniformly sampled frames, aiming to fit long video sequences into limited context length without retraining. The goal is to improve performance on video-language benchmarks using frozen, image-trained VLMs like LLaVA-1.6.

**Audience:**

Yes

**Claims And Evidence:**

Yes

**Requested Changes:**

1. **Reformulate the problem motivation**
Rather than improving image-only VLMs for video understanding — which lacks practical and conceptual grounding — the work could focus on **token compression for video-trained VLMs**, where managing the number of video tokens is a real and pressing challenge. This would make the method relevant to state-of-the-art models and realistic deployment scenarios.
2. **Conduct more rigorous ablation and architectural comparisons**
Include comparisons of key input designs (e.g., sampling+concat vs. sampling+grid) and analyze how compression choices affect temporal coverage and content preservation.
3. **Clarify technical descriptions and definitions**
Better define terms like “number of frames per thumbnail image,” and describe thumbnail construction, sampling intervals, and token allocation strategies in more concrete terms.
4. **Refocus experimental insights**
Go beyond surface-level trends and explore why the method works (or fails) under different configurations, and what this implies about the underlying model’s capacity and limitations.

**Strengths And Weaknesses:**

### Strengths
1. **Low-cost baseline for understanding image-VLM behavior on video tasks**. This paper provides a training-free baseline that helps analyze how image-only VLMs behave when naïvely extended to video. This may be of interest for diagnostic or pedagogical purposes in the study of modality gaps.
2. **Simple and reproducible methodology**
The proposed compression scheme — combining thumbnail images and uniformly sampled frames — is lightweight, easy to implement, and does not require access to model internals. This makes it suitable for plug-and-play experimentation on frozen image-based VLMs.
3. **Empirical consistency on select tasks**
While overall performance remains low, the method produces consistent (albeit modest) improvements over raw uniform sampling across several video-language benchmarks, indicating that certain structural token arrangements can help extract marginally more from frozen image models.

### Weakness

The central issue lies in a misformulated problem setting.

The central setting — improving video understanding via token compression on a training-free, image-only VLM — has **limited conceptual validity**. Video understanding requires models capable of capturing temporal dynamics, modeling long-range dependencies, and attending to fine-grained temporal cues — capacities that image-only VLMs inherently lack. In such a setup, no amount of token selection or compression can meaningfully overcome **the inherent limitations of an image-only VLM**. The problem, as posed, **focuses on optimizing inputs to a model that is structurally incapable of performing the task in a meaningful way**.

This mismatch is empirically evident. In Table 5, TS-LLaVA consistently underperforms PLLaVA, a minimal video-trained model, despite extensive compression design. Gains over naïve sampling are modest and task-dependent. Moreover, performance deteriorates when more thumbnails are added (Figure 4), indicating saturation effects. These patterns suggest that the bottleneck lies not in token quantity but in the image-only VLM’s inability to capture temporal structure.

Notably, the general motivation — reducing token count for efficient video understanding — is an important and active research direction, especially in practical settings where video inputs can quickly exceed model context limits. However, **this paper tackles the challenge under an unrealistic constraint — that the model cannot be trained on video data at all — and applies it to a model setting where token budget is not the limiting factor, and where no real video understanding is possible in the first place**. As a result, while the problem appears relevant on the surface, the formulation renders the solution ineffective and uninformative for advancing video-language modeling.

In addition, there are several secondary weaknesses.

1. **Lack of meaningful ablation and architectural justification**
The paper proposes a hybrid compression method using both thumbnail frames and sampled frames, but the architectural choices lack justification. For example, while Table 1 shows that grid layout is slightly better than concat layout, there is no comparison between the more critical combinations such as **sampling+concat vs. sampling+grid**. This omission weakens the claim that the proposed composition is optimal or even necessary. The method seems under-explored and arbitrarily defined.
2. **Superficial experimental interpretation**
The discussion section largely reports surface-level observations without deeper insight. For example, although Figure 6 shows that increasing the number of thumbnails harms performance, the authors offer no explanation or analysis of why this occurs or what it reveals about the method’s limitations. This reduces the scientific value of the paper and makes it difficult to draw transferable conclusions.
3. **Unclear or potentially misleading metrics**
Metrics such as “sampled token compression rate” are introduced but never clearly motivated. These ratios may not reflect real-world constraints or trade-offs, and their interpretation is not connected to performance. The metric naming is also confusing — it does not measure overall compression but rather the proportion of tokens allocated to one component, which may not be the primary concern in practice.
4. **Limited absolute performance**
Even when relative improvements are demonstrated (e.g., on VideoMME), the **absolute performance is far below existing open video models**, including relatively small recent models like Qwen2-VL. This makes it hard to argue that the method offers a practical advantage or will scale meaningfully beyond toy settings.
5. **Ambiguities in method description**
Some technical terms are poorly defined. For instance, “number of frames per thumbnail image” appears to refer to the temporal span compressed into a single image, but this is not clearly explained and could confuse readers. Similarly, how the thumbnails are actually generated (beyond uniform sampling) is underspecified.

---

> ### Author Response · Authors · 2025-08-30
> **Response to Reviewer oJ2u (Part 1)**
>
> Thank you for acknowledge the **Low-cost** nature of our method for video understanding, as well the **simplicity** and the **empirical consistancy**.
>
> ---
> > ### Critiques on task/problem setting
>
> We appreciate the reviewer’s concern regarding the training-free problem setting. However, we emphasize that training-free video LLMs are a well-established and actively studied paradigm [1,2,3,4], and our work builds directly on this line of research. This setting is not only valid but also highly valuable for several reasons:
> * While video understanding requires fine-grained temporal cues, frozen LLMs already possess strong sequential reasoning from language pretraining. Our goal, shared with prior work, is to structure compressed visual tokens so the LLM can effectively capture temporal dynamics. Crucially, our **training-free TS-LLaVA outperforms many training-based methods**, demonstrating that **effective design can compensate for the lack of training**.
> * Strong training-free designs often form the foundation for successful trained models. For example, we notice a recent extension to SF-LLaVA, SF-LLaVA1.5 [5]. SF-LLaVA1.5 uses the same compression strategy as SF-LLaVA, but trained with video-text paired data and achieves SOTA performance. While we currently lack resources to train TS-LLaVA end-to-end, our superior performance over SF-LLaVA in the training-free setting suggests strong potential for future extensions.
>
> ---
> > ### On architectural justification
>
> We focus on designing the best compression strategy for video LLMs, so we only evaluate at time the best compression strategy. The sampling+concat strategy has already been studied by previous works like FreeVA [3]. However, FreeVA is proven to be much less performant when comparing to SF-LLaVA.
> As a method outperforms SF-LLaVA by a large margin, we show that our strategy is much more effective.
>
> ---
> > ### On experimental interpretation
>
> Our interpretation is empirically grounded. Regarding Figure 4 (we note there is no Figure 6 in the paper, we assume the reviewer meant Figure 4), we state in Section 4.3 that temporal understanding is particularly critical for datasets like EgoSchema. Since our sampling pathway is designed to capture fine-grained temporal information, performance on EgoSchema is sensitive to the number of thumbnail images. The thumbnail images provide complementary global context but the thumbnail tokens replace the equivalent number of the sampled visual tokens. This trade-off highlights the importance of balancing detailed temporal sampling with holistic scene summarization for effective video understanding.
>
> ---
> > ### Clarification on metrics definition and method description
>
> * Compression rate: As defined on page 4 above Table 1, ``The compression rate, calculated as the ratio of visual tokens before to after compression``. We also provide a calculation process in Sec.4.3. This metric is introduced to give readers an intuitive understanding of the extent of token reduction in each method.
> * Thumbnail image: The construction of thumbnail images is described in Sec.3.2 (page 5): ``equidistant frames from all input frames to create a grid-view image (Grid)`` The Grid layout itself is clearly defined in Sec.3.1. Together with Figure 2 and 3, we believe this provides a comprehensive and visually supported explanation of how thumbnail images are generated.

---

> > ### Author Response · Authors · 2025-08-30
> > **Response to Reviewer oJ2u (Part 2)**
> >
> > ---
> > > ### Absolute performance with video LLM
> >
> > * We emphasize that **our training-free design already outperforms many existing video LLMs**, demonstrating the effectiveness of our compression strategy. It is important to note **that absolute performance on video benchmarks is highly dependent on the choice of backbone model**.
> > * Using video-trained MLLMs are not considered in our design, since we work on training-free methods. But here we also present results of our method combined with the model like Qwen2-VL [6] as requested. And we compare the results to a long-video tuned model LongVU [7].
> >
> >
> > **Performance comparison on VideoMME**
> > | Method | Visual Tokens | Short | Medium | Long | Overall |
> > |---|---|---|---|---|---|
> > | Qwen2-VL | 34560 | 64.89 | 50.56 | 45.89 | 53.78|
> > | LongVU | 4320 | 65.00 | 52.44 | 47.11 | 54.85 |
> > | TS-LLaVA (Ours) | 4320 | 66.89 | 52.33 | 45.33 | 54.85 |
> >
> > Even without long-video SFT, our method can reach the same level as LongVU. And with our compression method, with 12.5% visual tokens, we manage to maintain more accurate information for the model, which leads to improved performance as compared to the baseline Qwen2-VL.
> >
> > ---
> > *Reference*:
> >
> > [1] Wonkyun Kim, Changin Choi, Wonseok Lee, and Wonjong Rhee. (2024) An Image Grid Can Be Worth a Video: Zero-shot Video Question Answering Using a VLM.
> >
> > [2] Mingze Xu, Mingfei Gao, Zhe Gan, Hong-You Chen, Zhengfeng Lai, Haiming Gang, Kai Kang, and Afshin Dehghan. (2024) SlowFast-LLaVA: A Strong Training-Free Baseline for Video Large Language Models.
> >
> > [3] Wenhao Wu. (2024) FreeVA: Offline MLLM as Training-Free Video Assistant
> >
> > [4] Kai Han, Jianyuan Guo, Yehui Tang, Wei He, Enhua Wu and Yunhe Wang. (2024) Free Video-LLM: Prompt-guided Visual Perception for Efficient Training-free Video LLMs.
> >
> > [5] Mingze Xu, Mingfei Gao, Shiyu Li, Jiasen Lu, Zhe Gan, Zhengfeng Lai, Meng Cao, Kai Kang, Yinfei Yang and Afshin Dehghan. (2025) SlowFast-LLaVA-1.5: A Family of Token-Efficient Video Large Language Models for Long-Form Video Understanding.
> >
> > [6] Peng Wang, Shuai Bai, Sinan Tan, Shijie Wang, Zhihao Fan, Jinze Bai, Keqin Chen, Xuejing Liu, Jialin Wang, Wenbin Ge, Yang Fan, Kai Dang, Mengfei Du, Xuancheng Ren, Rui Men, Dayiheng Liu, Chang Zhou, Jingren Zhou and Junyang Lin. (2024) Qwen2-VL: Enhancing Vision-Language Model's Perception of the World at Any Resolution.
> >
> > [7] Xiaoqian Shen, Yunyang Xiong, Changsheng Zhao, Lemeng Wu, Jun Chen, Chenchen Zhu, Zechun Liu, Fanyi Xiao, Balakrishnan Varadarajan, Florian Bordes, Zhuang Liu, Hu Xu, Hyunwoo J. Kim, Bilge Soran, Raghuraman Krishnamoorthi, Mohamed Elhoseiny and Vikas Chandra. (2024) LongVU: Spatiotemporal Adaptive Compression for Long Video-Language Understanding.

---

### Review · Reviewer_u9U5 · 2025-08-18

**Summary Of Contributions:**

This paper presents a thumbnail-and-sampling approach for training-free video large language models, termed TS-LLaVA. The method demonstrates improved performance under the same token budget across multiple video benchmarks. In addition, the paper provides a comparison of various pooling strategies. Overall, TS-LLaVA is a concise and effective contribution; however, the techniques employed have already been explored in prior literature.

**Audience:**

Yes

**Broader Impact Concerns:**

This work could be applied to some other computation-intensive topics, e.g., video-based scene understanding.

**Claims And Evidence:**

Yes

**Requested Changes:**

The technical novelty appears limited, and there is substantial overlap with existing methods in the literature. I encourage the authors to clarify and highlight the unique aspects of their approach compared to prior work.

**Strengths And Weaknesses:**

## Strengths

- **Simplicity and Effectiveness:**
  The proposed TS-LLaVA method is straightforward to implement, relying on a thumbnail-and-sampling approach to compress video frames for input to a pre-trained image LLM. This simplicity is a merit, as it enables easy adoption without the need for complex model modifications or additional training data. The authors demonstrate that such a simple strategy can be highly effective, underscoring the practical value of leveraging existing image-based LLMs for video tasks.

- **Empirical Performance Gains:**
  The paper provides comprehensive experimental results across several video understanding benchmarks, such as MVBench and MLVU. Notably, TS-LLaVA achieves state-of-the-art performance among training-free video LLMs, and even surpasses GPT-4V on the MVBench benchmark with the 34B model. This is compelling evidence that the proposed method is not only conceptually appealing but also empirically competitive, especially considering it does not require additional video-text paired training data.

- **Systematic Comparison of Pooling Strategies:**
  The authors compare the efficacy of different pooling and token compression strategies, helping clarify why their particular approach works well. This comparative analysis adds to the paper’s rigor and helps justify the design choices.

## Weaknesses

- **Lack of Novelty Relative to Prior Work:**
  While the thumbnail-and-sampling strategy is effective, similar techniques for temporal frame sampling and grid-based thumbnail construction have already been explored in prior works, such as IG-VLM. The paper does not clearly articulate what unique insights or improvements TS-LLaVA brings over these existing methods. For example, the process of selecting equidistant frames and forming a grid-like thumbnail image has precedent, but the manuscript does not systematically analyze or highlight any significant technical differences or advantages over these previous approaches.

- **Limited Technical Depth Beyond the Sampling Strategy:**
  Other than the thumbnail-and-sampling approach for visual token compression, the paper does not introduce additional technical innovations. The method largely depends on the capabilities of existing pre-trained image LLMs and does not propose new model architectures, loss functions, or training paradigms. As a result, the contribution may be perceived as incremental, especially for readers already familiar with prior literature on video frame sampling and video LLMs.

---

> ### Author Response · Authors · 2025-08-30
> **Response to Reviewer u9U5**
>
> Thank you for acknowledge the **simplicity and effectiveness** of our method, as well as the **systematic comparison of pooling strategies** which is part of the motivations for our core method design. We are also encouraged to see your comments on the **strong performance gain** in empirical studies.
>
> ---
> > ### Concerns on novelty and technical depth
>
> We understand your concerns. However, the significance of our research extends beyond merely introducing a ”novel” method, which, we argue, is not the criterion for impact in a TMLR paper. Our contributions are substantial, and the findings are insightful:
> * We propose a **simple yet effective** compression strategy for video LLMs, achieving strong performance across multiple benchmarks. This shows that high performance can be achieved with a training-free approach, **challenging the necessity of compute-intensive training-based methods and setting a strong baseline for future research**.
> * We also offer deep insights for developing video LLMs.
>     * IG-VLM’s Grid layout struggles with long-term relationships due to limited frames. We extend it to Grids, which still hampers fine-grained action recognition from reduced resolution. In contrast, TS-LLaVA, leveraging sampled tokens, handles these tasks far more effectively.
>     * As also noted in Sec. 4.4, SF-LLaVA uses more frames in fast pathway, where similar adjacent frames lead to less distinctive features after aggressive pooling, harming understanding. In contrast, TS-LLaVA samples tokens from higher frame rates and complements them with summarized thumbnail tokens, preserving more video information.
>     * TS-LLaVA is empirically grounded and delivers significantly better performance. **Designing proper compression strategy for video LLMs is a challenging and non-trivial task**.

---

### Decision · Action_Editor_ssgg · 2025-10-06

**Recommendation:** Reject

**Additional Comments:**

In this submission, the authors presented a method for training-free video large language models (LLMs). Specifically, based on the idea of scarcity of high-quality, curated video-text paired data, and the similarity between images and videos, the authors proposed to extend image LLMs for video understanding tasks, by constructing visual tokens through a thumbnail-and-sampling strategy. Experimental results on benchmarks show reasonable performance of the proposed method on benchmark datasets.

This paper was reviewed by three expert reviewers. The authors provided detailed responses to the review comments, and the reviewers subsequently reviewed these responses. The final recommendations among the three reviewers leaned towards a negative decision, with 1 Reject, 1 Leaning Accept and 1 Leaning Reject. The main concerns raised by the reviewers are related to the lack of sufficient evidence supporting the key methodological claims, unclear distinction from prior approaches, and limited demonstrated contribution beyond existing work. The impact and the un-well-justified problem setting are another part of the main concerns. The reviewers agreed on the active and attractive research topic this paper studied and the performance achieved. While the topic is of interest to the TMLR audience and the results are encouraging, the submission does not yet provide a clear or well-supported case for its claimed advances.

As a result, the AE regrets to inform the authors that this paper, in its current form, is not ready for publication in TMLR. The authors are encouraged to carefully consider the comments and suggestions from the reviewers to strengthen their paper for a future submission.

**Audience:**

Yes

**Audience Explanation:**

The studied topic of video LLMs and training-free methods would be of interest to some individuals in TMLR's audience.

**Claims And Evidence:**

No

**Claims Explanation:**

Some of the claims made in the submission are supported by convincing and clear evidence. But there are some overclaims of the proposed new method, which was not supported by accurate, convincing evidence. Particularly, the proposed main idea has been explored in previous related works, but the paper did not present clear evidence of the differences and the claim about the "...the novel Thumbnail-and-Sampling compression strategy". The claims about extending image LLMs for video understanding are not well supported and justified as well.